# Impact of *N*-Acetyl Cysteine (NAC) on Tuberculosis (TB) Patients—A Systematic Review

**DOI:** 10.3390/antiox11112298

**Published:** 2022-11-21

**Authors:** Daniel Adon Mapamba, Elingarami Sauli, Lucy Mrema, Julieth Lalashowi, David Magombola, Joram Buza, Willyhelmina Olomi, Robert S. Wallis, Nyanda Elias Ntinginya

**Affiliations:** 1National Institute for Medical Research-Mbeya Medical Research Center, Mbeya 53107, Tanzania; 2The Nelson Mandela African Institution of Science and Technology, Arusha 23118, Tanzania; 3The Aurum Institute, 535 Park Avenue, New York, NY 10065, USA

**Keywords:** tuberculosis, *N*-Acetyl Cysteine (NAC), antioxidant, glutathione, lung health, culture conversion, inflammatory cytokines

## Abstract

Sustained TB infection overproduces reactive oxygen species (ROS) as a host defense mechanism. Research shows ROS is destructive to lung tissue. Glutathione (GSH) neutralizes ROS, although it is consumed. NAC is a precursor of GSH synthesis, and administering an appropriate dose of NAC to patients with respiratory conditions may enhance lung recovery and replenish GSH. The present review searched for articles reporting on the effects of NAC in TB treatment from 1960 to 31 May 2022. The PICO search strategy was used in Google Scholar, PubMed, SciFinder, and Wiley online library databases. The COVIDENCE tool was used to delete inappropriate content. We eventually discovered five clinical trials, one case report, seven reviews, in vitro research, and four experimental animal studies from the twenty-four accepted articles. The use of NAC resulted in increased GSH levels, decreased treatment time, and was safe with minimal adverse events. However, the evidence is currently insufficient to estimate the overall effects of NAC, thus the study warrants more NAC clinical trials to demonstrate its effects in TB treatment.

## 1. Introduction

Tuberculosis (TB) remains a leading cause of morbidity and mortality in people with pulmonary and extra-pulmonary forms of the disease, and has only recently been surpassed by the COVID-19 pandemic [1]. The pandemic has compromised the TB control program, including both human and financial resources for the fight against TB, as the results dropped between 2019 and 2021 [1]. Despite well-established and widely-publicized methods of prevention and treatment, the TB disease still affects people all over the world. The main causes of its persistence include HIV infection and poverty. The World Health Organization (WHO) estimated that there were 10.6 million incident TB cases and 1.6 million fatalities in 2021 [1]. Additionally, it has been observed that 85% of reported cases of TB are successfully treated [2]. Despite an apparent microbiologic cure, studies have shown that TB patients with a follow-up of at least 24 months still have some sort of persistent pulmonary impairment [3]. According to Ravimohan et al., (2018), pulmonary tuberculosis is a risk factor for long-term lung impairment and yet is frequently overlooked despite its high prevalence and link to a lower quality of life [4].

Sustained TB infection elicits an immunological process whereby the cellular immune response by alveolar macrophages, dendritic cells, and neutrophils as first-line defending cells acts on M. tb by engulfing, processing, and presenting peptides to naïve T cells that differentiate into T helper 1 cells (Th1), which undergo maturation and produce interferon gamma (IFN-γ), and activate macrophages and neutrophils to produce anti-microbial peptides including the reactive oxygen species (ROS), tumor necrosis alpha (TNF-α), and induced nitrogen species (iNOS) as part of the defense mechanism [5,6]. However, excessive production of ROS destroys lung tissues through the oxidation process [7,8].

Glutathione (GSH) is a non-protein thiol consisting of a tripeptide of glycine, cysteine, and glutamic acid, which are produced naturally by body cells. GSH has been found to be a free radical scavenger [9]. GSH has also been shown to protect against ROS and reactive nitrogen species by neutralizing free radicals and converting them into oxidized glutathione (GSSG) and water (2GSH + O-GSSG + H_2_O), a reaction that consumes GSH [10,11,12]. Cysteine amino acid as a precursor to GSH synthesis is required. Although it is considered conditionally essential, human synthetic capacity and dietary content are also limited.

NAC is a synthetic form of cysteine, a sulfur-containing amino acid that is temperature stable, a supplement required to replenish GSH [13]. It has been used routinely in the treatment of acetaminophen toxicity, which protects against fatal liver injuries [14]. Several studies have been conducted to evaluate and identify the use of NAC as a drug supplement in the treatment of several diseases, including the recent COVID-19 pandemic. NAC is a mucolytic and antioxidant drug is found to reduce the viscosity of the sputum, clean the bronchi, help to relieve dyspnea, and improve lung function [15,16]. Researchers went further to evaluate the effects of NAC against microbial activity and anti-TB drug effects during and after TB treatment [17,18,19,20]. The current review looked at study designs, methods, countries of origin, benefits for TB patients, Adverse events, and immunological responses. The review’s second goal was to identify the in vivo and in vitro study designs conducted to assess the effects of NAC on TB disease progression, including treatment reduction, sputum culture conversion rate, lung function, and GSH expression.

### 1.1. What Is Already Known on This Topic?

NAC serves as a precursor for GSH biosynthesis. The product was approved by the WHO for the treatment of acetaminophen intoxication. Studies are researching the product’s potential to treat a number of illnesses, such as the recent COVID-19 pandemic, TB, COPD, chronic bronchitis, and others.

### 1.2. What Does This Present Study Add?

The present review has found that few studies have been conducted to find out the effects of NAC in TB treatment and therefore highlights the importance of conducting more clinical trials on the use of NAC in TB treatment in order to demonstrate the general effect of adjunctive NAC in TB treatment.

## 2. Materials and Methods

The present systematic review was conducted by using a systematical literature search for all available NAC articles in the databases. Search terms were created following the population, concepts, and context (PCC) format. We categorized our review title into two parts, whereas the–“TB patients’ treatment” part of the research title was categorized as population, “the effect of adjunctive NAC” part was categorized as an intervention. The first group comprised search terms related to population targeting:

“tuberculosis treatment”[All Fields] OR “lung cavitation”[All Fields] OR “lung exacerbations” [All Fields] OR “COPD”[All Fields] OR “Respiratory problems”[All Fields] OR “lung impairment”[All Fields] OR “lung disorder” [All Fields] OR “interferon gamma”[MeSH] OR “inflammatory cytokines”[All Fields] OR “ROS”[All Fields] OR “Reactive oxygen-species”[All Fields] OR “lung inflammation”[All Fields] OR “Tuberculosis-disease”[All Fields] OR “lung-TB”[All Fields] OR “Standard anti-TB” OR “TB regimen”[All Fields] OR “relapse TB”[All Fields] OR “host-directed therapy”[All Fields].

The Cysteine group comprised the intervention concepts for the TB disease. The search terms were:

“GSH”[All Fields] AND “pulmonary-TB”[All Fields] OR (“Antioxidant”[All Fields] OR “Glutathione”[All Fields]) OR “pulmonary tuberculosis”[All Fields] OR “COPD”[All Fields] AND “TB”[All Fields] OR “pulmonary-tb”[All Fields] OR “Cysteine”[All Fields] OR “NAC”[All Fields] OR “acetyl-cysteine”[All Fields] OR “Precursor”[All Fields] OR “*n*-acetyl cysteine”[All Fields] OR “*N*-acetyl-cysteine”[All Fields] OR “Sulfur-containing”[All Fields] OR “Amino-acid”[All Fields] OR (“adjunct”[All Fields] OR “TB-treatment”[All Fields] OR “adjunctive”[All Fields] OR “replenish-GSH”[All Fields] OR “NAC impact”[All Fields] OR “NAC effects”[All Fields] OR “NAC-supplement”[All Fields]) OR “non-thiol”[All Fields] OR “Cysteine”[All Fields] OR “*N*-Acetyl cysteine”[All Fields] OR “Glycine”[All Fields] OR “cysteine”[MeSH Terms] OR “glutamine”[All Fields] OR “glutamic acid”[All Fields] OR “Hepatotoxicity”[All Fields] OR “DiLI”[All Fields].

We conducted a search of the identified terms in PubMed, Google Scholar, SciFinder, and Wiley online library databases. Each group of search terms were identified either in Abstracts, Free Full Text, full text, books and documents, case reports, clinical studies, clinical trials, in phase 1–4, clinical trial protocols, comparative studies, Meta-Analysis, observational studies, randomized controlled trials, research support, US Government, Reviews and systematic Reviews, and human subjects. We identified 42,205 results in the first group (as a population) and 113,106 results from the second group (as an intervention). Further searching using #1AND#2 was done by combining the results of the population and an intervention, as identified above. The initial database searches were performed on 12 April 2022, and updated on 31 May 2022. Twenty-four studies were selected for full-text review based on the abstracts obtained by this search approach. In addition, we looked through the reference lists of all the papers and reviews we obtained to see if there were any other studies that were relevant prior to data extraction. The articles included reported on clinical trials: randomized, double-blind or open-label, placebo-controlled, prospective, pilot, and case studies of oral, inhalation, and intravenous NAC supplementation taken on a regular basis during the course of TB treatment. We also included experimental studies investigating the in vitro effects of NAC on the harvested cells, either from humans or experimental animals, and studies conducted in the trial of NAC used in combination with other approved drugs. Review articles on the effects of NAC were included in this review as well see (Tables 1–4). We excluded trials of NAC conducted in other diseases apart from TB. See (Figure 1).

The present review work was not registered, and the protocol was not prepared.

The primary outcome measures of the review were to evaluate clinical trials on the effects of NAC on the progression of TB disease, such as improved lung function, sputum culture conversion, increased levels of GSH expression, and changes in cytokine production, specifically IL-10 and TNF-α. NAC supplement dosage and treatment duration were also assessed. The secondary outcome measures were to determine whether a protective effect or adverse event resulted from the course of NAC usage during and/or after TB treatment.

### Statistical Methods

Unadjusted meta-analyses were conducted for each intervention, comparison group, and outcome using a random effect model in STATA software (version 16, STATA Corporation, College Station, TX, 4905 Lakeway Drive, USA). Relative risk was used when the outcome was culture conversion, and risk difference was used when the outcomes were adverse events. *I*^2^ statistics were used to assess heterogeneity between studies. The values of *I*^2^ statistics greater than 50% represented high heterogeneity. For all analyses, the *p*-value for statistical significance was set at 0.05.

## 3. Results

After searching 37 citations and 8035 full-text papers, and by removing the duplicates and irrelevant articles using the Covidence tool, 207 articles were assessed for eligibility. Finally, 24 articles were included, of which 1 article was not yet published but rather was an accepted manuscript. The articles were searched from 1960, when NAC use began, until 2022. Of the identified and included articles, 100% were disseminated after 2004, 53.8% of the articles were published between 2016 and 2020, and 19.2% of the articles were already published from 2021 to 31st May 2022. Most of the articles were reported from North America (34.8%) and (13.04%) reported from Brazil or Iran. Furthermore, 8.7% of the identified articles are from the United Kingdom, India, South Africa, and China separately. Five (20.8%) of the identified articles were clinical trials, seven (29.2%) were systematic literature reviews or in vitro study designs, four (16.7%) were experimental animal model designed studies, and one (4.2%) was a case report. Almost all articles found that the NAC replenished GSH, which boosted host immunological function and improved TB treatment, including lung function and reduced time to sputum culture positivity, with minimal or minor side effects; see (Figure 1) and (Table 1, Table 2, Table 3 and Table 4).

### 3.1. Data Charting

The data charting process was completed after data extraction conducted by the use of the online COVIDENCE program tool, which was freely offered to be used by the organization team. The tools automatically identify duplicates and ease the data extraction process.

Details of articles included in review (Table 1, Table 2, Table 3 and Table 4).

Table 1 presents the in vitro studies that were carried out to assess the effect of NAC on TB.

Table 2 presents studies using experimental animal models to examine the effect of NAC on TB.

Table 3 presents clinical trials and a case study that was done to assess the impact of NAC on TB.

Table 4 presents studies on NAC’s impact on TB.

### 3.2. NAC Effect on Sputum Culture Conversion

Understanding NAC’s mucolytic property and that it may have an effect on microbial activity, the review looked into studies that reported on the effect of NAC in TB, with a focus on sputum culture conversion. We found two studies from Mahakalkar et al. [17] and Safe I.P [19], which both looked into the effect of NAC on sputum culture conversion (Table 5). The forest plot from two studies on the use of NAC in TB is shown below.

Table 5 presents sputum culture conversion from two studies. The overall effect of the NAC intervention group was not significantly different from the control, with a pooled RR of 1.10 and a 95% CI of 0.98–1.23. As for the heterogeneity between the studies, since the *I*^2^ is 0%, this suggests no important heterogeneity was observed between studies. Although the Mahakalkar paper reported that in the NAC group, 23 patients achieved sputum negativity in three weeks, while 14 patients were in the PLACEBO group and Safe. P reported 11 (61.1%) patients in the NAC group had a negative culture at week 8, while only 7 (33.3%) were in the control group.

### 3.3. Adverse Events of NAC in TB Subjects

NAC has been investigated as a supplementary drug for patients and reduces hepatotoxicity. The present review identified mild adverse events that were reported by Izabella P. Safe et al. [31], including nausea, vomiting, and hepatotoxicity. However, the article reported that up to 30% patients were experiencing hepatotoxicity are living with HIV, and the infection appeared to be one of the risk factors. Baniasadi et al. [29] reported that NAC protects against drug-induced hepatotoxicity caused by the standard anti-TB drugs therapy (INH, RIF, and PZA combination) in elderly populations. Again, milder adverse events were reported in a case study by Fox et al [32], Oral NAC was associated with nausea and vomiting, suggesting further investigation in clinical trials being required. Three studies identified by Kranzer et al. [34] reported that NAC reduced ototoxicity in 146 patients with end-stage renal failure receiving aminoglycosides, while eighty-three studies described how the administration of NAC for more than 6 weeks increased abdominal pain, nausea and vomiting, diarrhea, and arthralgia in 1.4–2.2 times. The meta-analysis from three studies were conducted to evaluate the adverse events was represented by a forest plot below (Table 6).

Table 6 presents adverse events between the control and NAC groups from three studies. The results showed a risk difference between the two groups with a 95% CI of −0.03 (95% CI: −0.42–0.35), although there was significant inter-study heterogeneity.

### 3.4. Immunological Responses

A pilot study conducted by Guerra et al. examined the effects of NAC (alone and in combination with IL-2 and IL-12) on up-regulating NK cell cytotoxic receptors and determined the levels of GSH in NK cells derived from HIV-positive, as well as the survival of H37Rv M. tb strain in monocytes cultured in the presence of NK cells (both from HIV-positive individuals). Findings treating NK cells with IL-2, IL-12, and NAC, inhibited the growth of H37Rv. When combined with GSH, it improved the ability of NK cells to control M. tb infection. Lin et al [21]. discovered that NAC inhibits the expression of TNF-α and caspase-9 genes as well as the translation of apoptotic proteins. Also Venketaraman et al. (2008) [24] showed that NAC decreased the levels of IL-10, IL-6, TNF-α, and IL-1 in blood cultures derived from TB patients and also showed the efficient control of intracellular M. tb infection in blood cultures derived from healthy subjects compared to TB patients.

### 3.5. GSH Expression Levels

We also looked into the effect of NAC on GSH expression. Two studies conducted by Mahakalkar et al. (2017) [17] in a clinical trial to investigate the effect of NAC on the production of glutathione peroxidase, an enzyme that catalyzes the oxidation and dimerization of GSH. The finding was that 600 mg NAC taken twice a day significantly raised Glutathione Peroxidase levels. A clinical study by Safe. P. et al. [31] measured the effect of 600 mg NAC taken twice a day on the GSH expression study found GSH levels were elevated only in RIPENAC patients at 60 days compared to levels found at the baseline. Furthermore, GSH levels were higher in RIPENAC patients compared to the RIPE group at day 60. It has been observed that NAC enhanced GSH levels by 2-fold in the RIPENAC group compared to the control at 60 days of treatment. It is possible that restoring GSH may negatively impact other markers, as the generation of ROS has been proposed as a mechanism of action for both macrophages and TB drugs. The current study did not explain how this effect will translate to clinical outcomes.

### 3.6. Lung Function

TB disease impairs normal lung function, and the effect persists even after the microbiological cure [3]. In several studies on lung disease, spirometry is the gold standard for measuring airway obstruction [5]. Researchers have reported that half of TB survivors present with some form of lung impairment [3,40]. NAC is now used as a food supplement and some clinical trials have investigated the role played by NAC in improving lung function impaired during TB infection. Forced expiratory volume (FEV1) has been very extensively studied in this regard; even small decreases in FEV1 are associated with increased standardized mortality risks. Conversely, there is a known long-term FEV1 benefit and a possible short-term benefit from treatment shortening. It is also possible that only some patients will benefit from NAC. The present review did not identify any studies or research on the use of NAC in improving lung outcomes in TB patients.

## 4. Discussion

A meta-analysis was performed from five identified clinical trials following similarities in their objectives and/or variation in study outcomes between studies. Two studies looked into the effect of NAC on sputum culture conversion, three studies investigated adverse events associated with the use of NAC, and three of the studies also looked into the effect of NAC on GSH expression levels. The findings from these clinical trials were presented with a clinical meaning based on the outcomes. The importance of NAC usage in TB is clear, and this adds to the current knowledge in the field by demonstrating that NAC may be a relevant candidate for adjunct therapy in TB. However, there is a need for multicenter clinical trials to study the effects of NAC, long-term follow-up investigations of TB subjects, and studies of NAC in multi-drug resistant TB. The use of a higher NAC dose and an investigation of its effects on teenagers may be an advantageous move to advocate the usefulness of NAC as a drug supplement.

### 4.1. Mechanism of NAC

In vitro and in vivo data have shown that NAC can protect the lungs against toxic agents by increasing pulmonary defense mechanisms through its direct antioxidant properties and as a precursor of GSH synthesis, [41,42,43,44]. which is depleted in patients with M. tb. Both in vitro and in vivo experiments have demonstrated that oxidative stress, which is increased in M. tb infection, may contribute to pathological abnormalities and functional changes in the lungs [45]. Increased ROS are common in M. tb infection and have been shown to reduce the synthesis of elastin and collagen [46]. Increased levels of ROS may also increase IL-1, IL-8, prostaglandins, and leukotriene production in several cell systems [47]. Finally, oxidative stress activates the transcription factor nuclear factor kappa-beta (NF-kb), which switches on the genes for TNF-alpha and all other inflammatory proteins [5]. In vitro, NAC acts as a precursor for GSH because it can penetrate cells easily and is subsequently deacylated to deliver cysteine [48]. The sulfydryl group from cysteine neutralizes external and internal toxic agents, such as NO, cellular aerobic respiration, inflammatory cytokines, and the metabolism of phagocytes [48]. NAC has shown potential function in immunological function, reducing hepatotoxicity, improvement of depleted GSH, sputum culture conversion rate, and other functions; however, the mechanisms underlying therapeutic and clinical applications are unknown [49]. Most studies have reported milder adverse events when administered orally compared to other routes of administration, compared to I.V route of NAC administration oral routepresents with nausea, vomiting and abdominal pain, generally anti TB therapy have been reported with Drug Induced Hepatotoxicity (DIH) and Drug induced liver injury (DiLI) effect to the TB opatients, it is Plausible that adjunctive NAC may countereffect the resulted and reported A’E.

### 4.2. Effect of NAC on Microbial Activity

Sputum culture conversion is used as a microbiological end time point in clinical trials during TB treatment [49,50]. The most common factors that affect time to sputum culture conversion include multidrug resistance, HIV co-infection, previous alcohol drinking, high baseline sputum smear grading, number of resistant drugs at initiation, number of active drugs taken, and delaying treatment for more than one month [51,52]. Salindri et al. [53] also pointed out that diabetes affects time to culture conversion in newly diagnosed MDR-TB patients [53,54]. In a prospective, randomized, parallel group by Mahakalkar et al. [17] in vivo early sputum negativity achieved by the addition of NAC to the standard TB drugs in a prospective, randomized, and parallel fashion was reported. Safe P et al. [31] found that NAC was associated with faster sputum negativity, improved radiological response, and improvement in weight. The rate of change of TTP has been proposed as a marker of drug antimicrobial activity. Regimes that show faster culture conversion values seem likely candidates for shorter durations. Shorter regimens will have a more rapid effect on these markers, but this is not well established.

Three studies used 400 mg/kg—which is equivalent to a 32.4 mg/kg human dose—of NAC in animal model design to study its effects on TB. These research findings provide evidence that the in vivo use of adjunctive NAC against TB with optimal dosage can affect the microbial activity of M. tb by boosting host immunological function, including the replenishment of GSH, or may directly affect M. tb growth [8,16,23,24]. The use of NAC in TB patients may impact the pathogen’s survival as well as improve lung function in the patient if the adjunctive NAC is taken at optimal dosage.

Blood samples from healthy and TB subjects were used in in vitro investigations by Venketaraman et al. [24] who found that H37Rv growth was nearly two times higher in healthy subjects. In contrast, the H37Rv strain’s growth was inhibited when blood cultures were treated with NAC (10 mM). However, the benefits of NAC in inducing the growth control of H37Rv were reversed when buthionine sulphoximine (BSO) (500 mM) was used. BSO is a sulfoximine derivative that suppresses glutathione levels [41,43]. The compound inhibits gamma-glutamylcysteine synthetase, the enzyme needed for the initial stage of glutathione biosynthesis. The discovery of the compound is a significant new technique for highlighting the activity of glutathione in vivo [55,56]. The experiment demonstrated the significant role of NAC in both the regulation of M. tb proliferation and the expression of GSH. However, there is insufficient data to support the antioxidants NAC and GSH’s in vivo effects.

## 5. Limitations of the Review Process

The present review was limited with few searched articles related to the question, and it was difficult to conclude and recommend the effectiveness of NAC to the pathogen as well as to the host.

## 6. Conclusions

Our review could not come to the best conclusion about the effect of NAC on TB treatment due to the limited literature available on the use of NAC in TB subjects. Therefore, this review warrants more clinical trials on the use of NAC to demonstrate the effects of NAC. The ongoing TB-SEQUEL-NAC study in Tanzania, ClinicalTrials.gov Identifier: NCT 03702738, aims to evaluate clinical trials on the effect of NAC in TB treatment using a higher dose of 1200 mg BID of NAC in patients with TB, with and without HIV co-infection. It is plausible that the study would add evidence to support the use of NAC as a drug supplement in TB treatment.

## 7. Recommendation

Our review shows that NAC has a modest but significant effect on increased intracellular GSH expression rates and decreased inflammatory cytokines in TB patients and thus improves post-TB lung disease. However, due to limited clinical trials on NAC, this present review therefore points to the need for more clinical studies on the use of NAC during TB treatment or the treatment of other respiratory diseases.

## Figures and Tables

**Figure 1 antioxidants-11-02298-f001:**
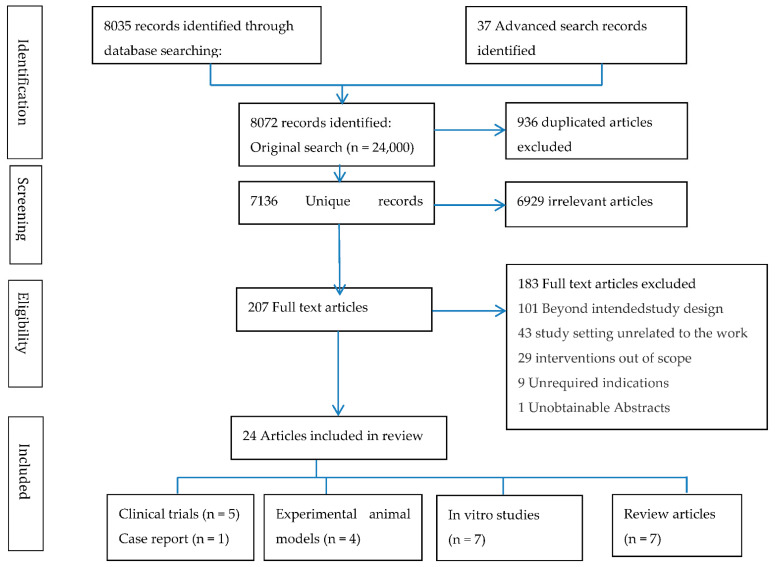
PRISMA flowchart of study selection process.

**Table 1 antioxidants-11-02298-t001:** In vitro studies.

Author and Time of Publication	In Vitro Experiment	Objectives	Intervention (Methods Overview)	Results/Outcome	Reference
Lin (2020)	Cell culture experiment, derived from Experimental animal model	RAW264.7 macrophages were used to explore the immunological response and cell damage of host cells after incubation with Mycolic Acid.	NAC conc. of 600 mg/mL for 2 h was used to treat incubated cells.	NAC inhibited the expression of the TNF-α and caspase-9 genes and reduced the translation of apoptotic proteins.	[21]
Amaral (2016)	70 (32) Plasma from 30 TB, subjects 20 LTBI, and 20 Healthy control	NAC directly impairs the growth of several species of M.tb in vitro independent of its inhibitory effects on the host NADPH oxidase System.	10 mM NAC was used to treat M.tb infected Macrophages from 70 subjects.	NAC significantly decreased ROS accumulation, lipid peroxidation, and DNA oxidation, while restoring cell viability.	[20]
Coleman (2004)	Monocyte cells derived from blood collected from 24 Healthy volunteers (21–24)	Use of GSH, NAC, Lipoic acid (LA), dihydrolipoic acid (DHLA; to modulate the toxicity of (N1-4-*N*,*N*-dimethylamino-1-naphthylidene) pyridine-2-carboxamidrazone (Compound 1), and Isoniazid (INH), which has demonstrated both Mononuclear leukocyte toxicity and anti-mycobacterial action.	Isoniazid (INH) and Compound 1-treated cells were incubated with either NAC, GSH, DHLA, or LA in 1 mM; final concentration.	GSH and NAC showed abolition of Isoniazid (INH) toxicity to the mononuclear leukocyte cells	[22]
Venketaraman (2006)	20 Healthy and TB subjects	Study examined the role of GSH in immunity against TB infection in samples derived from healthy and HIV-human subjects.	10 mM NAC conc. was used in cell culture experiment to find the relationship between GSH levels and the ability to kill intracellular M. tb.	NAC resulted in more efficient control of intracellular M. tb infection in blood cultures derived from healthy subjects compared to TB Patients; NAC treatment caused down-regulation of the synthesis of IL-1, IL-6, and TNF-α.	[23]
Venketaraman (2008)	12 Healthy and TB subjects	To determine the extent to which GSH levels are decreased in patients with active TB and examine the relationship between GSH and the ability to kill intracellular M.tb and other host immune functions, such as cytokine production.	The effect of 10 mM NAC/BSO in altering the intracellular survival of H37Rv M.tb strain.	NAC decreased the levels of IL-10, IL-6, TNF-a, and IL-1 in blood cultures derived from TB patients and also showed efficient control of intracellular M. tb infection in blood cultures derived from healthy subjects compared to TB patients	[24]
Teskey (2018)	13 subjects (20–65)	Examined the effect of NAC on M. tb infection.	NAC 10 mM + antibiotic in altering the survival of M. tb, Elevated GSH levels.	NAC results in significant reduction of M. tb burden in both healthy and diabetic individuals.	[25]
Khameneh (2016)	NA	Investigation of the antibacterial activity of vitamin C and NAC individually and in combination with RIF and INH against different strains of *S. aureus* and M.tb.	MIC on cell cultures NAC (final conc. 40.0 mg/L) or vitamin C (final conc. 40.0 mg/L).	Combination of vitamin C and NAC was able to reduce the hepatotoxicity of the anti-tb drugs and enhanced antimicrobial activity.	[26]
Guerra (2012)	23 subjects	The study demonstrated the treatment of NK cells with IL-2 + IL-12 + NAC resulted in inhibition in the growth of H37Rv M.tb strain.	20 mM NAC was used in treatment with IL-2 + IL-12.	Results unveil an important pathway by which cytokines in conjunction with GSH, enhanced the functions of NK cells to control M. tb infection.	[27]

**Table 2 antioxidants-11-02298-t002:** Studies on experimental animal Models.

Author And Publication Time	Animal Species	Study Design	Objectives	NAC Dose	Results/Outcome	Reference
Amaral (2016)	C57BL/6 Micen = 20	NAC directly impairs the growth of several species of mycobacteria in vitro independent of its inhibitory effects on the host NADPH oxidase system. This anti-mycobacterial effect was also observed in an experimental model in vivo.	NAC exerts anti mycobacterial activity in vivo.	400 mg/Kg daily for 6 days.	Lung burden of M. tb-infected mice decreased by 0.5 log10 at day 7 compared to untreated mice.	[20]
Palanisamy (2011)	Guinea pigs were used	To establish the presence of oxidative stress conditions during experimental TB in guinea pigs and to determine whether antioxidant therapy could reverse the adverse effects of progressive inflammation, including lessening the bacterial burden and disease severity.	Guinea pigs of 9 Month of age were aerosolized using the Madison infection chamber with H37Rv M.tb with a conc. Of 10^6^ CFU/mL, followed vaccination with or without BCG treated with or without NAC 400 mg/kg.	NAC conc. Of 400 mg/kg was used to treat infected mice.	Daily administration of NAC resulted in nearly one log reduction in the number of bacilli in the spleen on day 30. No significant differences in the numbers of bacilli were observed between control and NAC-treated groups on days 30 and 60 in lungs and peribronchial lymph node. An increase in whole blood GSH was seen in NAC-treated animals compared to the mock-treated control group on day 60 of infection.	[28]
Lin (2020)	Cell culture and Experimental animal model (ICR mice)	Animal experiments were performed to investigate the role of NAC in antagonizing the effects of Mycolic Acid in the induction of apoptosis and autophagy.	NAC on Mycolic Acid; ICR mice were used to evaluate the lung injury.	The intranasal NAC dose used in the studies is not mentioned in the study.	NAC inhibited the expression of the TNF-α and caspase-9 genes and reduced the translation of apoptotic proteins. NAC reduced the secretion of IL-6 significantly; also, NAC attenuated apoptosis and autophagy in response to incubation with Mycolic acid.	[21]
Vilchèze (2021)	Cell culture and experimental animal model (CBA/J mice)	Assessing the function of NAC in vitro, in boosting activity with various combinations of first- and second-line TB drugs against drug-susceptible and multidrug-resistant M. tuberculosis strains.	Adjunctive activity of NAC combined with first- or second-line TB drugs in cultures of M. tb, in M. tb-infected macrophage and in M. tb-infected mice.	NAC, 0.5 or 1 g/kg; was used orally to treat infected CBA/J mice.	NAC enhanced the killing of M. tb by first- and second-line TB drugs in vitro.	[18]

**Table 3 antioxidants-11-02298-t003:** Clinical trials.

Author and Publication Time	Country	No. of Subjects (Mean, Age)	Clinical Criteria (Aim of the Study)	Length of Research (Months)	Intervention (Methods Overview)	Results/Outcome	Reference
Baniasadi (2010)	IRAN	60 (60)	Protective effect of NAC against anti-TB drug-induced hepatotoxicity.	Over 2 weeks	NAC (600 mg, orally, BID	NAC protects against anti-TB drug-induced hepatotoxicity.	[29]
Moosa (2021)	SOUTH AFRICA	102 (38)	Assessing whether i.v NAC hastens liver recovery in hospitalized adult patients with anti-tuberculosis drug induced liver injury (AT-DILI).	Not reported	NAC dosage was as per Acetaminophen toxicity dosage 150 mg/kg over 1 h, 50 mg/kg over 4 h, and 100 mg/kg over 16 h	NAC did not shorten time to ALT < 100 U/L in subjects with AT-DILI, but significantly reduced length of hospital stay. [nausea and vomiting, anaphylaxis, pain at drip site.]	[30]
Safe (2020)	BRAZIL	39 (≥18)	Impact of adjunctive NAC treatment on host immune response and redox homeostasis in population of hospitalized patients with HIV-associated TB.	16 months	NAC 600 mg BID for 8 weeks	RIPENAC group had elevated plasma levels of GSH compared to RIPE group at the same time-point.	[31]
Safe (2020)	BRAZIL	39 (≥18)	Testing the hypothesis that NAC is safe, well tolerated and secondarily efficacious as adjunctive anti-TB therapy in hospitalized individuals with HIV-TB.	16 months	NAC 600 mg bid for 8 weeks	The use of NAC in the HIV/TB population seems promising in terms of safety, and mycobacterial clearance results indicate that RIPE plus NAC regimen is suitable for a larger phase III trial.	[19]
Mahakalkar (2017)	INDIA 67(18–60)		Effect of NAC (add-on to DOTS Category I regimen) on sputum conversion, radiological improvement, and GSH peroxidase; weight and immunological response compared to placebo.	18 Months	Standard anti-TB treatment with or without NAC600 mg daily	Adjunctive NAC increased GSH peroxidase levels in TB patients. GSH increase might reduce ROS, TNF-α production. The combination of NAC effects on both the pathogen and the host might be required to observe Early sputum negativity also Radiological improvement (87.5% was achieved by NAC group compared to 33.33% in placebo).	[17]
Fox (2020)	USA	1 (30)	Reversal of ALF due to DILI in a patient receiving anti-tubercular agents for active TB. NAC be considered for patients with anti-TB-associated DILI.	Not reported	I.V. NAC dosage for acute acetaminophen was followed by infusions of 50 mg/kg over 4 h and 100 mg/kg over 16 h, as well as 100 mg/kg as a continuous for a period of 48 h until 2 additional bags had been infused.	Oral NAC reported with nausea and vomiting NAC use can be considered for patients with anti-TB therapy-associated DILI.	[32]

**Table 4 antioxidants-11-02298-t004:** Literature reviews.

Author and Publication Time	Objectives	NAC Dose	Results/Outcome	Reference
Atkuri (2007)	Summarizes the biochemical and pharmacological aspects of NAC that make it a “wise choice” to treat cysteine/GSH deficiencies.	NAC ≥ 600 mg/day	Compared to the placebo group, a small fraction of individuals to whom oral NAC was administered experienced nausea, vomiting, and heartburn.	[33]
Kranzer (2015)	Provision of evidence for the safety and oto- protective effect of NAC when co-administered with aminoglycoside in MDR-TB.	NAC (600 mg, orally, twice daily Co-administered with aminoglycoside	NAC reduced ototoxicity in 146 patients with end-stage renal failure receiving aminoglycosides, while 83 studies reported with an increased mild adverse events.	[34]
Mokhtari (2017)	The paper presents a review on various applications of NAC in treatment of several diseases.	Use of NAC in the treatment of several diseases	NAC is a safe and well-tolerated supplementary drug without any considerable side effects.	[35]
Dawit (2020)	NAC-attenuate hearing loss in MDR-TB	NAC (600 mg, orally, twice daily	NAC appears to have various beneficial effects on TB treatment.	[36]
Young (2020)	The review discusses promising pre-clinical candidates and forerunning compounds at advanced stages of clinical investigation in TB host-directed therapeutic (HDT) efficacy trials.	TB preventative therapy. HDTs could enhance anti-mycobacterial properties of lung phagocytes, which would prevent infection. For TB contacts and LTBI, we host strengthening preventative strategies, including vitamin supplementation, NAC, and BCG re-vaccination.	Applicability of HDTs to MDR-TB, TB treatment shortening, TB/HIV, and TB-derived lung diseases, although highlighted in some studies, have not been considered for all HDTs.	[37]
Schwalfenberg. (2021)	Review the clinical usefulness of NAC as treatment or adjunctive therapy in a number of medical conditions.	NAC 1200 mg/day	NAC appears to be well tolerated with minimal side effects when used as a supplement or in treatment of various disorders.	[38]
Tenório (2021)	Overview of the medicinal effects and applications of NAC to human health based on current therapeutic evidence.	NAC 1200 mg/day	There is a need to clarifying adequate dosages and treatment protocols.	[39]

**Table 5 antioxidants-11-02298-t005:** Forest plot on sputum culture conversion.

				Risk Ratio	Risk Ratio
Study or Subgroup	Log [Risk Ratio]	SE	Weight	IV, Random, 95% CI	IV, Random, 95% CI
Mhakalkar (2017) [17]	0.086	0.06	97.3%	1.09 [0.97, 1.23]	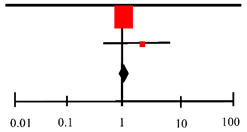
Safe (2020) [19]	0.3	0.36	2.7%	1.35 [0.67, 2.73]	
**Total (95%CI)**			**100.0%**	**1.10 [0.98, 1.23]**	
Heterogeneity: Tau^2^ = 0.00; Chi^2^ = 0.34, df = 1 (*p* < 0.56); *I*^2^ = 0%					
Test for overall effect: Z = 1.55 (*p* = 0.12)					Favors [Experimental] Favors [Control]

**Table 6 antioxidants-11-02298-t006:** Forest plot on adverse events.

				Risk Difference	Risk Difference
Study or Subgroup	Risk Difference	SE	Weight	IV, Random, 95% CI	IV, Random, 95% CI
Baniasadi (2010) [29]	−0.38	0.08	34.8%	−0.38 [−0.54–0.22]	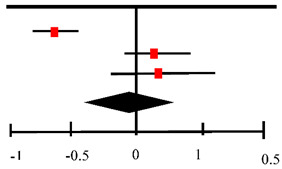
Moosa (2020) [30]	0.14	0.09	34.2%	0.14 [−0.04, 0.32]	
Safe (2020) [31]	0.16	0.14	31.0%	0.16 [−0.11, 0.43]	
**Total (95% CI)**			**100.0%**	**−0.03 [−0.42, 0.35]**	
Heterogeneity: Tau ^2^ = 0.10; Chi^2^ = 22.81, df = 2 (*p* < 0.0001); *I*^2^ = 91%					

## Data Availability

The data presented in this study are available in the article.

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
