# Peer review of "Impact of N-Acetyl Cysteine (NAC) on Tuberculosis (TB) Patients—A Systematic Review"

_antioxidants, 2022, doi:10.3390/antiox11112298_

Round 1

Reviewer 1 Report

Summary: this is an excellent review on on NAC in TB, breaking down the results by in vitro, animal model, human studies and PKPD, antimicrobial factors & mechanisms of action. It is well done, and I have no major criticisms. The following are some provocative questions I wonder if the authors might consider addressing in their discussion and conclusion. 

Is time to sputum conversion important? For example, if a theoretical drug were to increase the time to positivity by 7 days (7 days slower), but improve long-term lung health (improved 6MWT or PFTs), wouldn’t that still be worth it? Is TTP a good marker of TB treatment success or should we be focused on the individuals long term function? I suspect the latter and wonder if the authors might comment in the discussion. (In fact, a recent metformin study did show improved radiographic lung healing at the expense of a few days slower sputum culture conversion. Mechanistically, metformin could be slowing ROS production by decreasing electron leak from the mitochondria inner membrane. Would metformin or everolimus theoretically therefore be synergistic with NAC?  Do all TB patients need NAC? Andrade et al demonstrated two TB sub-groups (one HO1 hi and the other MMP1 high. Maybe the HO1 group represents the TB endotype that would most benefit from NAC? Maybe only TB patients with low Trolox or high MDA need NAC, and the TB patients with high reducing capacity have little benefit? Maybe give a few sentences on how NAC should or should not limit requisite intracellular signaling driven by NAC to discuss if it could or could not be harmful if given in excess? 

Minor suggestions: 

Line 13: is it confirmed that administering NAC enhances lung recovery and replenishes GSH? This finds like a conclusion rather than an introduction. Can it be worded so it is clear this is an introduction and not a conclusion? 

Line 31-32: NAC replenishes consumed GSH and neutralizes ROS. Is this in vitro or in vivo animal or human or all? Please clarify and add some citations. (Ref 7-8 appropriate?)

Line 58-59 needs a reference. ROS is critical for intracellular signaling, but excessive ROS damages lung tissue. Maybe consider adding this caveat and the need for homeostatic regulation. 

Before going to publication, Table 1 “in vitro” studies is extremely small and hidden. Increase the size so this is obvious these are all in vitro studies. (line 213; Same for Table 2- line 215). Table 1 is an excellent brief review. Consider working on the cosmetics a bit before publication (extra spaces in certain places)

Baniasadi 2010: how is drug induced hepatotoxicity defined? 

Safe2020: No other outcomes were measured other than GSH levels? Change in radiography? TTP? Nothing else? 

Mahakalk 2017: were PFts or 6MWT measured? How was radiography assessed? 

Line 239-242 needs references

Line 392: the typical reviewer does not know BSO inhibits GSH; kindly remind them

Author Response

Dear Reviewer, 1

Please see the attachment in the box, as the responses to your comments have been uploaded.

Thanks. 

Reviewer 2 Report

Unfortunately, in general, the information in the manuscript is rather chaotic, and not precise enough, there are many repetitions and information unrelated to the aim of the work.

1. The conclusion presented in the abstract does not correspond to the aim of the work and the conclusion presented in the main text.

2. In the section "0. What is already known on this topic?" there are inaccurate, incorrect, ambiguous statements.

3. The most important result of the work or the most important conclusion must be presented in the section "What does this present study adds?" But not the aim of work.

4. In the section "1. Introduction" lines 43-59 provide redundant general information.

5. In several places: lines 136-143, 183-192, 222-236 the same information is repeated.

6. Figure 1. It is not clear what "Wrong" means. How is it possible that so many peer-reviewed publications dedicated to the same topic have been "Wrong"? The term "Wrong" should be defined or several examples of "Wrong" should be given.

7. The information presented in Tables 1-4 is disorganized, without any system.

8. Results section, "3.2. NAC effect on sputum culture conversion“. The information in lines 238-249 would be appropriate for the discussion section (in the right context), but not for the results section.

9. Results section, „3.6. Mechanism of NAC”. The information in lines 307-333 would also fit the discussion section (in the right context), but not the results section.

10. Statements in lines 336-356, and 359-365 contain information unrelated to the aim of the work.

11. The discussion part is too short. Doesn't sound scientific discussion.

12. The conclusion is not specific, unclear.

Author Response

Dear Reviewer, 2

Please see the attachment containing responses to your comments.

Thanks.

Round 2

Reviewer 2 Report

None